# Lipids, Tetraspanins, and Exosomes: Cell Factors in *Orthoflavivirus* Replication and Propagation

**DOI:** 10.3390/v17101321

**Published:** 2025-09-29

**Authors:** Magda L. Benitez-Vega, Carlos D. Cordero-Rivera, Jose De Jesus Bravo-Silva, Ricardo Jimenez-Camacho, Carlos Noe Farfan-Morales, Jonathan Hernández-Castillo, Marcos Pérez-García, Rosa M. del Ángel

**Affiliations:** 1Department of Infectomics and Molecular Pathogenesis, Center for Research and Advanced Studies, Mexico City 07360, Mexico; magda.benitez@cinvestav.mx (M.L.B.-V.); carlos.cordero@cinvestav.mx (C.D.C.-R.); jose.bravo@cinvestav.mx (J.D.J.B.-S.); ricardo.jimenez@cinvestav.mx (R.J.-C.); cfarfan@cua.uam.mx (C.N.F.-M.); jonathan.hernandez@cinvestav.mx (J.H.-C.); marcos.perezg@cinvestav.mx (M.P.-G.); 2General Direction, National Polytechnic Institute (IPN), Mexico City 07738, Mexico; 3Department of Natural Sciences, Metropolitan Autonomous University (UAM), Cuajimalpa Campus, Mexico City 05348, Mexico

**Keywords:** *Orthoflavivirus*, lipids, tetraspanin, tetraspanin-enriched microdomains, extracellular vesicles, replicative cycle, antiviral strategies

## Abstract

The cellular membrane is a dynamic structure composed of lipids and proteins organized into specialized domains that facilitate interactions between extracellular molecules and the intracellular environment. Tetraspanins are a family of transmembrane proteins involved in diverse cellular processes, including membrane stabilization and fusion, endocytosis, extracellular vesicle formation, and the organization of proteins and lipids at specific membrane sites known as Tetraspanin-Enriched Microdomains (TEMs). These lipid–protein interactions play a critical role in the replicative cycle of *Orthoflavivirus*, including dengue, Zika, and West Nile, by facilitating viral entry, replication, assembly, and egress. In addition, tetraspanins also regulate the biogenesis and function of extracellular vesicles, contributing to viral dissemination, persistent infection, and immune evasion. This review summarizes the current knowledge on the structural and functional aspects of tetraspanins, their interplay with lipids, and their emerging roles in the *Orthoflavivirus* replicative cycle. We also discuss how these insights may inform the development of antiviral strategies targeting membrane organization and virus–host interactions.

## 1. Introduction

*Orthoflavivirus* is a genus that includes a diverse group of medically necessary human pathogens posing serious public health challenges, particularly in Latin America, Asia, and Africa. These viruses are associated with a broad range of clinical manifestations, from mild febrile illness to severe conditions such as encephalitis, hemorrhagic fever, and congenital malformations, considerably impacting healthcare systems and socioeconomic structures [1,2]. Clinically relevant members of this genus include yellow fever virus (YFV), West Nile virus (WNV), Japanese encephalitis virus (JEV), and Kyasanur Forest disease virus (KFDV), among others [3]. Currently, dengue virus (DENV) and Zika virus (ZIKV) stand out as the most significant orthoflaviviruses species due to their widespread circulation and major impact on public health.

DENV has shown an alarming resurgence in recent years, particularly in Latin America, which is currently experiencing the largest outbreak. ZIKV, although with fewer reported cases in recent years, remains of great concern due to its association with severe neurological complications and congenital anomalies.

ZIKV has caused major outbreaks with devastating consequences, including congenital Zika syndrome malformations and neurological disorders such as Guillain-Barré syndrome in adults [4,5]. Meanwhile, DENV is projected to have caused over 5 million infection cases in 2023, with estimates surpassing 12 million cases in 2024, and presents a wide range of clinical spectrum, ranging from asymptomatic infections to life-threatening disease presentations, from asymptomatic infections to severe, potentially fatal disease forms [1,6]. Despite the substantial disease burden caused by these viruses, no specific antiviral therapies or universally effective vaccines are currently available, underscoring the urgent need for novel therapeutic approaches.

In this context, host cell factors, including lipids, tetraspanins (Tspan), and extracellular vesicles (EVs), have emerged as key players in the *Orthoflavivirus* replication cycle. These cellular components are involved in multiple stages of the viral life cycle, such as entry, replication, and egress, and offer attractive targets for developing antiviral strategies. This review focuses on the role of these host-derived elements in *Orthoflavivirus* biology, highlighting replication and propagation, emphasizing their relevance for the design and optimization of therapeutic interventions, with potential for developing or enhancing antiviral strategies.

## 2. Lipid Involvement in *Orthoflavivirus* Replicative Cycle

The replication cycle of orthoflaviviruses is intricately dependent on the host cell’s lipid metabolism, placing lipids as essential elements for viral propagation and attractive targets for therapeutic intervention [7,8,9]. These enveloped viruses possess a single-stranded, positive-sense RNA genome protected by a membrane primarily derived from the host endoplasmic reticulum (ER) [3,10].

During the entry process, the envelope lipids in the viral particle, particularly phospholipids such as phosphatidylserine and phosphatidylethanolamine, play a key role in mediating interactions with cellular receptors, promoting membrane binding and subsequent fusion [11,12]. Once internalized, viral replication occurs within ER-derived membranous replication complexes, which are highly enriched in cholesterol and other lipids. These specialized microenvironments support efficient viral RNA synthesis and protein translation [9,13]. At later stages, the association between the nucleocapsid and ER-derived membranes and lipid droplets has been critical for virion assembly and maturation [14,15,16].

Given this dependence on host-derived membranes and lipid stores, orthoflaviviruses have evolved strategies to manipulate cellular signaling pathways that regulate lipid metabolism. Among these, the PI3K/Akt/mTOR signaling pathway regulates lipid metabolism and the autophagy pathway, showing complex interactions during orthoflaviviruses infection. In the case of ZIKV, it is reported that the viral infection inhibits the mTORC1 signaling, enhancing autophagy, but obstructing protein synthesis. However, the authors demonstrate a strong correlation between autophagy and viral RNA transcription [17]. Another study identifies that nonstructural proteins NS4A and NS4B have been shown to inhibit the PI3K/Akt pathway, preventing the activation of mTORC1 and promoting autophagy [18]. Similarly, in DENV, NS4A induces autophagy via a PI3K-dependent process [19]. Furthermore, DENV NS1 has been reported to stimulate autophagy through interaction of the LKB1–AMPK axis, suppressing mTOR activity [20]. Other orthoflaviviruses, such as JEV, have also been shown to inhibit mTORC1, leading to autophagy induction [21]. Also, the NS1 and NS2A proteins of JEV obstruct the interaction between AMPK and LKB1, impeding the AMPK activation and increasing lipid synthesis [22]. These findings demonstrate that the PI3K/Akt/mTOR pathway to control autophagy is a common strategy among orthoflaviviruses and is essential to their replication cycle.

Autophagy is a fundamental cellular process through which various intracellular components, including damaged organelles and pathogen-derived material in the context of infection, are degraded within lysosomal vesicles [23]. Based on the route by which cargo is delivered to lysosomes, autophagy is classified into macroautophagy (canonical), the most extensively studied during DENV and ZIKV infections, as well as microautophagy and chaperone-mediated autophagy (CMA); in addition, several non-canonical macroautophagy routes have been described [23]. Nevertheless, DENV and ZIKV have been shown to hijack this process to facilitate their replication cycle; inhibiting key canonical autophagy components significantly reduces infection by these flaviviruses [24,25,26]. Notably, autophagy regulates lipid metabolism to support viral replication; this selective process, lipophagy, breaks down lipid droplets through triglyceride hydrolysis to release fatty acids that fuel β-oxidation and ATP production during infection [27,28]. At the same time, the PI3K/Akt/mTOR pathway is known to play an essential role in lipid metabolism by regulating both catabolic and anabolic processes. In particular, mTORC1 promotes de novo lipogenesis by activating sterol regulatory element-binding proteins (SREBPs), thereby inducing the transcription of genes involved in fatty acid and cholesterol biosynthesis, and causing the accumulation of lipid droplets during infection [28]. These findings suggest these viruses have the capacity to modulate PI3K/Akt/mTOR pathway activity, highlighting a complex regulatory balance between autophagy and lipid metabolism (see Figure 1) [9,21,28].

This regulation may impact lipid-rich microdomains on the cell membrane, such as lipid rafts and tetraspanin-enriched microdomains, which serve as organizational platforms that concentrate receptors and intracellular signaling proteins, facilitating key processes like viral entry and assembly [29,30,31]. Among these, tetraspanins (Tspan) are particularly notable due to their ability to organize lipid domains and regulate multiple biological processes, including extracellular vesicle (EV) biogenesis [29,32,33,34].

Beyond their structural roles, Tspan molecules contribute to several stages of the orthoflaviviruses’ cycle. Their involvement in EV formation positions them as potential mediators of viral dissemination, expanding our understanding of how host factors support orthoflaviviruses’ spread and persistence [35,36].

## 3. Tetraspanin-Enriched Microdomains

Given their central role in lipid remodeling and viral dissemination, it becomes essential to understand how tetraspanins organize specialized plasma membrane structures known as tetraspanin-enriched microdomains (TEMs). Beyond passive barriers, membranes function as dynamic platforms supporting intracellular communication and initiating intracellular signaling pathways. In this context, membrane organization is critical, as compartmentalization into lipid-enriched microdomains facilitates the assembly, localization, and propagation of signaling complexes, ultimately influencing a wide range of cellular processes [37].

Lipid rafts are nanoscale, dynamic membrane assemblies enriched in sphingolipids, cholesterol, and specific proteins, which can transiently form and occasionally coalesce into larger, stabilized platforms [38]. The high content of sphingolipids and cholesterol reduces membrane fluidity, creating an optimal environment for the recruitment and clustering of signaling proteins and receptors [39,40,41].

In contrast, tetraspanin-enriched microdomains (TEMs) are structurally and functionally distinct from lipid rafts. They differ in molecular composition and exhibit distinct sensitivities to temperature, cholesterol levels, detergents, and protein palmitoylation [29]. Central to TEMs are Tspan proteins, which mediate specific homophilic and heterophilic protein–protein interactions (see Figure 2). These primary interactions give rise to an extended network of secondary associations, with protein palmitoylation contributing to the stability and organization of the domain [29].

Tspans are the principal structural components required to organize these multimolecular membrane complexes. Although they are typically not bound to external ligands, Tspan proteins orchestrate the assembly of TEMs involved in a wide range of cellular processes and signaling pathways [29,42,43].

Notably, lipid rafts and TEMs have been implicated in the internalization and replication of enveloped viruses, including members of the *Flaviviridae* family [44,45,46]. These specialized membrane domains provide a spatially organized environment that supports key events in the viral life cycle, such as receptor binding, endocytosis, trafficking, viral replication, nuclear export, and viral budding [46].

## 4. TEMs in the Replicative Cycle

The first step in the viral replicative cycle is the attachment of the virion to the host cell surface via one or more receptors or other membrane-anchored molecules [47,48]. Accordingly, viruses must engage with host cell surface components—such as receptors, proteases, and signaling molecules, many of which are located within membrane microdomains. Each mechanism of viral entry involves a distinct set of interactions between the virus and components of the cell membrane system [44,48,49].

Enveloped viruses, which possess a lipid membrane enveloping their nucleocapsid, are characterized by having structural viral proteins anchored to the envelope [47,48]. These proteins facilitate interactions with host cells, enabling viral entry and genome delivery through fusion of the viral membrane with the plasma membrane or intracellular compartments [50].

Tspan are transmembrane proteins enriched in membrane microdomains and involved in various cellular functions, including membrane stabilization and fusion, endocytosis, and the biogenesis of EVs [29,31]. Increasing evidence suggests that Tspans also participate in multiple stages of viral infection [44]. Specifically, Tspan has been proposed to facilitate viral replication by mediating viral attachment and entry, endosomal fusion, virion budding, and producing EVs that carry infectious viral material [44,45,46].

## 5. Tetraspanins

Tspan are a conserved family of small transmembrane proteins that play key roles in organizing specialized regions of the plasma membrane. These dynamic platforms facilitate the spatial coordination of membrane proteins, signaling molecules, and cytoskeletal components involved in various cellular processes, including adhesion, migration, endocytosis, and vesicle trafficking [29,51]. Many Tspan associate with membrane proteins and other Tspan, including themselves [31], forming a complex interaction network that can influence membrane remodeling, such as curvature, stiffness, and tension, thereby affecting multiple cellular functions [52].

Tspans are characterized by four transmembrane domains (TM1-TM4), connected by two extracellular loops, a small extracellular loop and a large extracellular loop (SEL and LEL), and short cytoplasmic N- and C-terminal tails (see Figure 3) [31,43,52]. The transmembrane domains are involved in intramolecular interactions and lateral associations that promote the assembly of TEMS [42,53]. Studies on CD9 have shown that TM1 and TM2 mediate intramolecular interactions within conserved regions that stabilize the protein core, while TM3 and TM4 contribute to intermolecular interactions, responsible for TEMs assembly [54]. Structural analysis has also demonstrated that transmembrane domains form an intramembrane cavity that can bind cholesterol molecules, suggesting that cholesterol may modulate Tspan function [55].

The LEL plays a central role in intermolecular interaction. It typically contains 69–132 amino acids and a conserved three α-helix core surrounding a hypervariable region. This region includes disulfide bridges between conserved cysteine residues, conserved glycosylation sites, and structural motifs such as β-strands or unstructured loops, depending on the specific Tspan [52,54]. The variable region of the LEL is implicated in homodimerization and specific interactions with partner proteins, playing a critical role in Tspan function [42].

Moreover, the SEL (13 to 31 amino acids) and LEL (69–132 amino acids) contribute to the specificity of Tspan-protein interactions. The LEL can be subdivided into a constant region of three α-helices and a variable region, which is often exposed at the end of the first helix near conserved glycosylation sites [31,44]. Between the second and third helices, depending on the type of Tspan, this region may include additional α-helices, β-strands, or unstructured loops [56]. Disulfide bridges between cysteine further stabilize the variable region. Additionally, Tspan often contains a tyrosine residue that targets cargo molecules to intracellular compartments [57].

The extracellular loops are also flanked by short N- and C-terminal cytoplasmic tails of (8 to 21 amino acids), oriented towards the cytosol [45,58]. Notably, the C-terminal residues of some Tspan, such as CD81 and CD151, contain PDZ-binding motifs that may mediate interactions with intracellular scaffolding proteins [59]. These cytoplasmic tails also contribute to cell signaling, serving as binding sites for molecules, such as protein kinase C and phosphatidyl inositol 4-kinase [60,61,62], thereby enhancing transmission of extracellular signals into intracellular signaling cascades [42,53].

The structural organization of Tspan, through intramolecular stabilization, lateral association with other proteins, and interaction with cytosolic partners, makes them attractive targets for pathogens like *Orthoflavivirus*. At the same time, these molecules represent promising candidates for the development of antiviral strategies aimed at disrupting critical steps of the viral replication cycle.

## 6. Tetraspanins in Viral Infections

Because of their broad biological functions, Tspan are exploited by numerous viruses at multiple stages of the replicative cycle, including attachment, internalization, replication, and egress [46,63]. Entry of human papillomavirus (HPV), influenza virus (IAV), Middle East respiratory syndrome coronavirus (MERS-CoV), and severe acute respiratory syndrome coronavirus 2 (SARS-CoV-2) all rely, directly or indirectly, on Tspan-mediated mechanisms [64,65,66,67,68]. Although the orthoflaviviruses’ replicative cycle and its host cofactors have been extensively characterized, the specific contribution of Tspan remains poorly defined.

*Orthoflavivirus* can enter cells through several routes involving molecules that act as attachment factors, such as proteoglycans, heparan sulfate, lectins, and integrins [49,69,70]. Depending on the receptor repertoire and cell type, entry proceeds either (I) by direct fusion of the viral envelope with the plasma membrane or (II) by receptor-mediated endocytosis followed by fusion within endosomes [71,72].

Some Tspan can act as bona fide viral receptors. The best studied example is CD81 in hepatitis C virus (HCV) entry. The viral E2 glycoprotein binds the large extracellular loop (LEL) of CD81, and antibody blockade of CD81 significantly reduces late entry events, indicating a role in endosomal trafficking [73,74]. Cholesterol-induced conformational changes in CD81’s LEL further modulate HCV entry efficiency [75]. CD81 also operates within a larger entry complex that includes epidermal growth factor receptor (EGFR), calpain-5 (CAPN5), and the E3 ubiquitin ligase CBLB [76].

By contrast, most orthoflavivirus-Tspan interactions are likely indirect. Tspan can also associate with cellular proteins that facilitate viral entry (Table 1), helping assemble or stabilize membrane microdomains supporting virus-receptor engagement and membrane fusion [62,77,78]. Precisely how TEMs and individuals, whether tetraspanins, orchestrate these events during orthoflaviviruses infection remains an open question that warrants further study.

Once the viral genome is released into the cytoplasm via fusion within the endosome, it is targeted to intracellular membranes to form replication complexes. These specialized membranous structures serve as platforms for viral RNA replication and translation [97].

Although tetraspanins are primarily localized at the plasma membrane, several studies have implicated them in later stages of infection, such as replication and translation, in viruses such as chikungunya virus (CHIKV), human immunodeficiency virus (HIV), and Venezuelan equine encephalitis virus (VEEV) [98,99,100,101]. In the case of the Flaviviridae family, emerging evidence also suggests that tetraspanins may influence post-entry events in the viral cycle.

For example, CD81 has been shown to correlate positively with HCV RNA replication. In HCV-infected or RNA-transfected cells with low CD81 expression, replication was reduced. However, early stages of RNA synthesis were not affected, suggesting a possible role for CD81 in the assembly or stability of the viral replicase complex [102]. Conversely, in HCV-expressing hepatoma cells, CD81 expression was downregulated, and CD81-knockout cells exhibited increased viral replication associated with enhanced pro-survival signaling, implying that CD81 may also contribute to viral persistence and chronic infection [103].

In the ZIKV context, infection alters the expression profile of several tetraspanins, including the upregulating CD63, which localizes to viral replication complexes. Notably, overexpression of CD63 in ZIKV-infected cells decreased capsid protein levels, suggesting a potential role in viral morphogenesis or assembly. Additionally, authors discuss a possible negative regulation of RNA-capside interaction by CD63 [104].

Although tetraspanin function has been less studied in mosquito cells, recent findings indicate its importance in arboviral replication and transmission. In DENV-2-infected mosquito cells, the Tspan Tsp29Fb is upregulated and interacts directly with the viral E protein. Functional studies using RNA silencing or antibody blocking of Tsp29Fb resulted in a significant reduction in viral replication and decreased viral content in extracellular vesicles, highlighting its involvement in virus propagation [105].

Another mosquito-tetraspanin C189 is strongly upregulated ~4-fold upon DENV2 infection in mosquito cells C6/36. Although C189 is not involved in virus attachment, it localizes to the plasma membrane and facilitates cell-to-cell viral spread [106]. This role was confirmed by knockdown experiments, where the reduction in C189 significantly impaired direct viral transmission between cells. This phenotype was rescued by co-transfection of a C189 expression vector, further supporting its role in *Orthoflavivirus* dissemination [107].

A recent study with tetraspanins from mosquito cells demonstrates the relevance of these proteins in the replication of orthoflaviviruses. The authors report the upregulation of CD151 in ZIKV and DENV2-infected cells C6/36, and the silencing of this Tspan led to a significant reduction in viral replication. Additionally, a co-immunoprecipitation essay revealed a direct interaction between CD151 and ZIKV NS2B and DENV2 capsid protein [108]. These evidence suggested that tetraspanins can interact directly with viral proteins to promote viral infection (see Table 2).

Tspans have also been implicated in viral egress in multiple viruses, including IAV, HIV, human cytomegalovirus (HCMV), and HSV-1 [64,98]. In Orthflavivirus, the late stages of replication involve the assembly of virions in the ER and trafficking through the secretory pathway [109]. Some Tspan, including CD63 and CD81, are enriched in late endosomes and multivesicular bodies (MVBs), compartments that overlap with the extracellular vesicle (EVs) biogenesis pathway [110]. These compartments are increasingly recognized as necessary for non-lytic viral release and intercellular transmission.

Recent evidence suggests orthoflaviviruses can hijack the EVs pathway to mediate cell-to-cell transmission. In this context, Tspans not only serve as structural components of EVs but also play an active role in organizing protein and lipid cargo [111].

Altogether, the involvement of Tspan in *Orthoflavivirus* infection is multifaceted and stage-specific. Their ability to interact with both membrane proteins and lipids positions them as central organizers of membrane microenvironments that orthoflaviviruses exploit during entry, replication, vesicular trafficking, and release (see Figure 4).

## 7. Tetraspanins in Extracellular Vesicles

Tspans also play essential roles in the biogenesis and function of EVs, particularly exosomes [58]. Growing evidence highlights their involvement in EVs biogenesis, cargo selection, target cell specificity, and uptake [58]. EVs are nanometric structures (30–150 nm) delimited by a lipid bilayer and present molecular components such as Tspan CD9, CD63, and CD81, which are commonly used as standard molecular markers for their identification and characterization [110]. Several reports indicate that Tspan functions are related to the regulation of EV functional properties, including uptake and fusion, as well as the regulation of EV composition. However, current research has produced contradictory results, keeping the understanding of the function of tetraspanins in EVs still unclear [112]. In addition to proteins, exosomes are also rich in sphingomyelin, gangliosides, phosphatidylserine, and cholesterol, but the relative abundance depends on the producer cell, the physiological stage, and the function of the exosome [113].

Exosomes have an endosomal origin, maturing into multivesicular bodies (MVBs) that contain intraluminal vesicles (ILVs), which result from the inward budding of the endosomal membrane [114]. These ILVs can be generated via endosomal sorting complex required for transport (ESCRT)-dependent machinery or ESCRT-independent pathways, such as the nSMase2-ceramide-dependent and tetraspanins pathways [115]. After the multivesicular bodies are formed, they are transported to the cell’s periphery, where the membrane of the MVBs and the cell membrane fuse, and the luminal vesicles are released into the extracellular space as exosomes [116]. Once the exosome reaches a target cell, it can interact through various mechanisms, including direct binding to membrane receptors, fusion with the cell membrane, or internalization via different routes such as phagocytosis, micropinocytosis, or lipid raft-, clathrin-, or caveolin-mediated endocytosis, to deliver its cargo [116]. These uptake pathways enable exosomes to deliver proteins, lipids, and nucleic acids between cells, participating in the intracellular communication, immune responses, and pathophysiological processes.

Due to their capacity to transfer biomolecules between cells, orthoflaviviruses have hijacked this pathway to promote viral dissemination, evade immune responses, and facilitate the spread of infection.

## 8. Extracellular Vesicles During *Orthoflavivirus* Infection

Over recent decades, knowledge about EVs has advanced significantly, particularly in the context of viral infections. EV biogenesis involves multiple complex biological steps that engage diverse molecular components and signaling pathways. Given the similarities between exosome biogenesis and viral replication—especially in the processes of assembly and egress—viruses can hijack the EV machinery to enhance viral propagation, manipulate host immunity, and modulate the cellular microenvironment. EVs can transmit infectious cargo as complete virions (single units or aggregates), individual viral components, or naked viral genomes. This packaging protects viral material from host enzymatic degradation and immune surveillance and confers attributes such as high biocompatibility, the ability to traverse biological barriers, and low immunogenicity [117,118].

In *Orthoflavivirus* research, several studies have demonstrated that EVs from infected cells carry both viral and cellular regulatory components, including proteins and RNA, that modulate host responses and facilitate infection in neighboring cells [119]. DENV and ZIKV have been the most extensively studied in vitro. For example, cells of Aedes aegypti and Aedes albopictus infected with DENV-2 or DENV-3 release exosomes containing full-length viral RNA and proteins (E and NS1). These vesicles can infect human and mosquito cells through interactions with the vector receptor Tsp29Fb, bypassing the need for classical virions, and establish an alternative virus-vector-host transmission route mediated by infectious vesicles [105]. Further characterization of exosomes derived from DENV-infected C6/36 cells revealed that these vesicles are morphologically larger than those from uninfected cells and contain virus-like particles. These exosomes can induce infection in naïve cells, supporting their function as independent transmission units, distinct from virions [120].

In the case of ZIKV, exosomes from infected C6/36 cells contain both viral RNA and E protein and can initiate infection in naïve mosquito cells, human monocytes, and endothelial cells. These EVs upregulate TNF-α and proinflammatory factors, such as TF and PAR-1, thereby increasing vascular permeability and suggesting a pivotal role in modulating immune responses and endothelial dysfunction during ZIKV infection [121]. Similarly, ZIKV-infected human monocytes secrete exosomes containing viral RNA and proteins (E and NS1), which can promote infection in naïve cells, stimulate monocyte differentiation and immune activation, and facilitate viral dissemination through mechanisms independent of classical virions, underscoring their importance in viral immunopathology [122]. Also, ZIKV exploits the EVs biogenesis machinery in Vero cells—particularly CD63-regulated pathways—to package viral RNA and structural proteins into exosomes. These vesicles are infectious, and their production is significantly reduced when CD63 is overexpressed, highlighting the central role of this tetraspanin in EV-mediated viral transmission [104]. In both ZIKV and DENV, the presence of virus-like particles within exosomes has been demonstrated by transmission electron microscopy, suggesting a novel transmission mode [120,121]. However, studies with ZIKV have shown that exosomes carry the E protein on their surface, attenuating antibody-dependent enhancement mediated by ZIKV E-specific and DENV-cross-reactive antibodies in cell culture and mouse models [123]. The entry route of the exosomes coated with the E protein could be by multiple pathways commonly used by EVs; however, whether the E protein is used as a ligand to trigger endocytosis remains unclear. Safadi et al. (2023) [124] further demonstrated that, during DENV and ZIKV infection, exosomes transport the NS1 protein, a key virulence factor, to uninfected cells, where they can trigger additional immunopathological responses. This association suggests a novel systemic viral toxicity dissemination mechanism via EVs (see Figure 5) [124].

Exosomes have also been implicated in the pathogenesis and spread of neurotropic *Orthoflavivirus* such as West Nile virus (WNV). For example, neurons infected with WNV release small EVs (sEVs) that contain viral genomic RNA and structural proteins, which can infect other neurons in vitro. This suggests an alternative route of viral spread that could contribute to neuroinvasion [35]. Another study demonstrated that WNV-infected human epithelial cells secrete sEVs with altered profiles of small noncoding RNAs, which activate antiviral immune responses in recipient cells, thereby reinforcing the emerging role of EVs in neurovirulence and immunomodulation during *Orthoflavivirus* infections [125].

Beyond in vitro studies, EVs also play a critical role in clinical samples of *Orthoflavivirus* infection. Vedpathak et al. (2023) found that platelet-derived EVs from dengue patients, particularly those with severe disease, disrupt vascular endothelial integrity by reducing junctional proteins such as Claudin-1 and Cadherin, while increasing inflammatory markers (CRP, SAA, sVCAM1, sICAM1), thereby contributing to vascular pathogenesis [126]. Kumari et al. (2023) reported that plasma EVs from patients with severe dengue carry cytokines and PD-L1, which suppress CD4+ T cell activation via the PD-L1/PD-1 axis, ultimately impairing the adaptive immune response [127]. Additionally, distinct EV microRNA profiles have been identified in the serum of patients with acute versus asymptomatic dengue, suggesting that EVs may serve as biomarkers for differentiating the clinical course of infection [128]. On the other hand, a study revealed that the exosomes of human primary dendritic cells infected with DENV3 contained several mRNAs and miRNAs related to immune responses, compared to the exosomes from non-infected cells. This result suggests that the differential immune modulation can be achieved through the EV pathway [129]. These results suggest that during DENV infection, the EV pathway can be exploited to favor viral viability, despite containing molecules that trigger immune mechanisms to counteract the viral infection.

These findings underscore the multifaceted role of exosomes in *Orthoflavivirus* infections. EVs contribute to viral transmission, immune evasion, inflammation, and disease pathogenesis, providing novel targets for biomarker discovery and developing antiviral therapeutics.

## 9. Therapeutic Targeting of Tetraspanins and Lipid Metabolism in *Orthoflavivirus* Infections

*Orthoflavivirus* infections pose a significant public health concern due to their global incidence [130,131]. In this context, tetraspanins (Tspan) have emerged as promising therapeutic targets requiring further investigation, given their role in various stages of the viral replication cycle and their involvement in critical cellular processes [34,132]. Experimental inhibition of Tspan—via RNA interference (RNAi) or monoclonal antibodies (mAbs)—has demonstrated antiviral effects in infections caused by IAV, HIV, HPV, coronaviruses, and orthoflaviviruses [34,132].

For example, RNAi silencing of CD81 or CD151 reduces HCV replication, while doxycycline-induced knockdown of CD9, CD63, and CD81 increases ZIKV replication; conversely, their overexpression limits infection [34,104]. Monoclonal antibodies targeting CD81 or CD151 have impaired HCV entry in vitro and conferred protection in humanized mouse models [34,133]. In HIV-1 models, anti-CD63 antibodies reduced macrophage infection more effectively than RNAi, while anti-CD9/CD81 antibodies modulated membrane fusion in lymphocytes.

Given their conservation and involvement in host pathways exploited by viruses, Tspan offer potential for antiviral therapies and broad-spectrum targets [34,132,133]. One promising approach is the indirect modulation of Tspan through cholesterol homeostasis. Tetraspanins like CD81, CD82, CD151, and CD63 interact closely with cholesterol, which is critical for their palmitoylation, membrane organization, and participation in tetraspanin-enriched microdomains (TEMs) [55,75,134,135,136,137,138]. Studies show CD81’s activity depends on its cholesterol-binding status: cholesterol-bound CD81 enhances HCV entry, while unbound CD81 is less functional [55,75,139].

This opens opportunities for repurposing lipid-lowering agents—statins, ezetimibe, and metformin—to disrupt cholesterol-rich microdomains and tetraspanin function. These drugs have demonstrated antiviral activity against DENV and ZIKV in vitro and in vivo, that have been associated with reduced viremia, decreased tissue viral loads, and attenuation of clinical disease severity [9,140,141,142,143]. Notably, statins such as atorvastatin and simvastatin exhibit more potent antiviral activity against HCV than lovastatin [144,145,146]. Since HCV belongs to the Flaviviridae family, manipulating host lipid metabolism may represent a potential therapeutic avenue for other flaviviruses [7,9]. Additionally, two novel compounds, isobavachalcone (IBC) and corosolic acid (CA), inhibit lipid synthesis by stimulating AMPK by binding to its active allosteric site, exhibit an inhibition of JEV infection, reducing viral loads in the brain, and mitigating histopathological alterations in an in vivo model [22]. Nevertheless, studies are needed to define the precise mechanisms and pharmacological targets involved, requiring further validation.

As mentioned in the previous section, orthoflaviviruses can exploit the EV pathway for dissemination, positioning EV biogenesis as another antiviral target. For example, the inhibitor GW4869, which blocks neutral sphingomyelinase-2 (nSMase2) and ceramide production, reduces exosome formation and impairs the replication of ZIKV, DENV2, WNV, and Langat virus (LGTV) in in vitro models [35,105,108,147,148]. However, a few in vivo studies have been performed using the GW4869 inhibitor. In vivo evaluation has only been performed in adult Ixodes scapularis ticks, where a reduction in LGTV dissemination was reported [149]. Although these findings were obtained in invertebrates, no reports are available from mammalian models to assess their potential as antivirals. Other inhibitors of ceramide metabolism, such as imipramine, display antiviral activity against ZIKV, WNV, and DENV in vitro but lack in vivo validation [9].

Similarly, DPTIP shows antiviral effects by interfering with viral entry and exit, inhibiting ZIKV and WNV replication through nSMase2 inhibition and lipid profile modulation [150,151]. While in vivo studies revealed a modest reduction in WNV viral load, transcriptomic analyses indicated an upregulation of anti-inflammatory genes, suggesting additional immunomodulatory benefits in neuroinflammatory contexts [152]. DPTIP remains the only inhibitor with demonstrated antiviral activity in a murine model. Despite these promising findings, knowledge about the toxicity of many of these compounds remains limited, as traditional murine model data are scarce. Notably, DPTIP remains the only inhibitor with demonstrated antiviral activity in a murine model. Despite these promising findings, knowledge about the toxicity of many of these compounds remains limited, as traditional murine model data are scarce.

Although the inhibition of EVs biogenesis and release has been investigated as a potential antiviral strategy (see Table 3), this approach warrants careful consideration. As mentioned, EVs play a dual role during infection, as they can transport both viral elements and host-derived molecules with antiviral properties. Consequently, future research should selectively target specific EV subpopulations that facilitate viral replication and dissemination, while conserving those involved in antiviral defense.

The broad-spectrum potential of host-directed antivirals offers a clear therapeutic advantage, as many viruses rely on conserved host pathways. Nevertheless, targeting these pathways does not eliminate the risk of resistance since viruses may adapt by exploiting redundant mechanisms or altering their interactions with host factors. Thus, although host-directed antivirals are generally less susceptible to classical resistance than direct-acting antivirals, the possibility of adaptive escape persists and should be carefully addressed in future therapeutic design.

Lipids, tetraspanins, and EV-targeted antiviral therapies represent promising strategies because they can avoid viral resistance and provide broad-spectrum activity. Though this approach also presents considerable challenges, as host pathways are essential for both viral replication and normal cellular physiology, their modulation is susceptible to off-target effects and toxicity. Moreover, the therapeutic window of many host-targeting agents may be narrow, requiring careful dose optimization to balance antiviral efficacy with the preservation of cellular homeostasis. However, these findings support a multifaceted antiviral strategy directed at tetraspanins, lipid metabolism, and EV biogenesis, three interconnected cellular processes exploited by orthoflaviviruses during their replicative cycle. Therefore, future research should focus on advancing the therapeutic validation of these strategies to demonstrate their potential as antivirals.

## 10. Conclusions

Orthoflaviviruses have evolved sophisticated mechanisms to exploit host cellular pathways, including tetraspanin-enriched membrane domains, lipid metabolism, and extracellular vesicle (EV) trafficking, to complete their replication cycles and enhance their transmission. The evidence reviewed here highlights the multifunctional role of tetraspanins in viral entry and replication, morphogenesis, and viral spread, particularly through EV-mediated pathways. The interplay between membrane remodeling, vesicle formation, and host immune modulation provides novel targets for therapeutic intervention.

By understanding how Orthoflaviviruses hijack these processes, especially in mammalian and mosquito cells, we can identify vulnerabilities in the viral life cycle that are amenable to pharmacological disruption. The combined targeting of tetraspanins, lipid metabolic pathways, and EV biogenesis offers a promising strategy for broad-spectrum antiviral development. Further studies are needed to elucidate the molecular mechanisms underlying these interactions and to explore their translational potential in clinical settings.

## Figures and Tables

**Figure 1 viruses-17-01321-f001:**
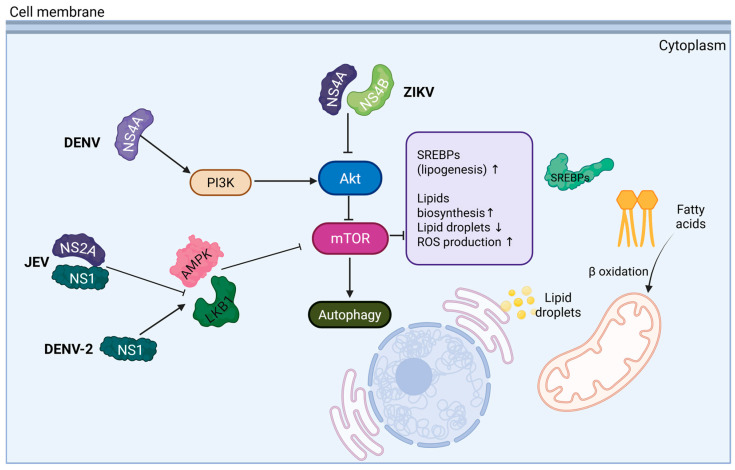
The PI3K/Akt/mTOR signaling pathway regulates autophagy and lipid metabolism, showing complex interactions during *Orthoflavivirus* infection. Nonstructural proteins from DENV (NS4A), DENV-2 (NS1), JEV (NS1, NS2A), and ZIKV (NS4A, NS4B) modulate distinct points of the PI3K/Akt/mTOR pathway, regulating processes such as autophagy and lipogenesis to support viral replication. The arrows indicate the activation of the signaling pathway. The inhibitor lines indicates the inhibition of the signaling pathway.

**Figure 2 viruses-17-01321-f002:**
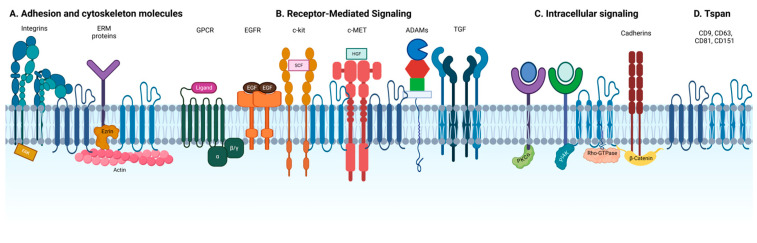
Tetraspanin-enriched microdomains. Schematic representation of TEMs. Tspan can model the plasma membrane interacting with (**A**) key adhesion molecules, ERM proteins, (**B**) signaling receptors), (**C**) intracellular proteins, (**D**) other tetraspanins and enzymes that constitute these microdomains.

**Figure 3 viruses-17-01321-f003:**
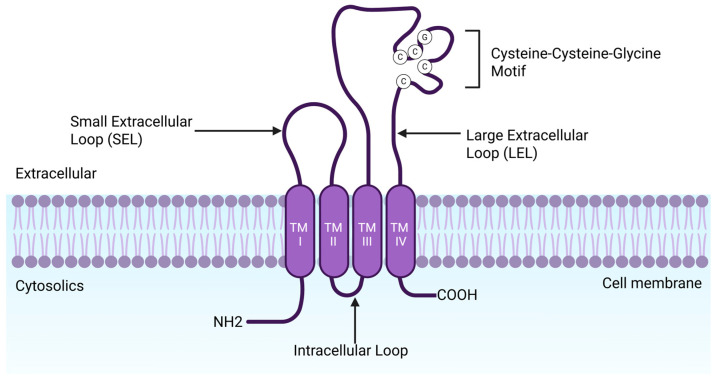
Tetraspanin structure model. Tspan includes four transmembrane domains (TM), two N-terminal and C-terminal cytoplasmic tails, and two extracellular regions (SEL and LEL). LEL also contains a conserved CCG sequence.

**Figure 4 viruses-17-01321-f004:**
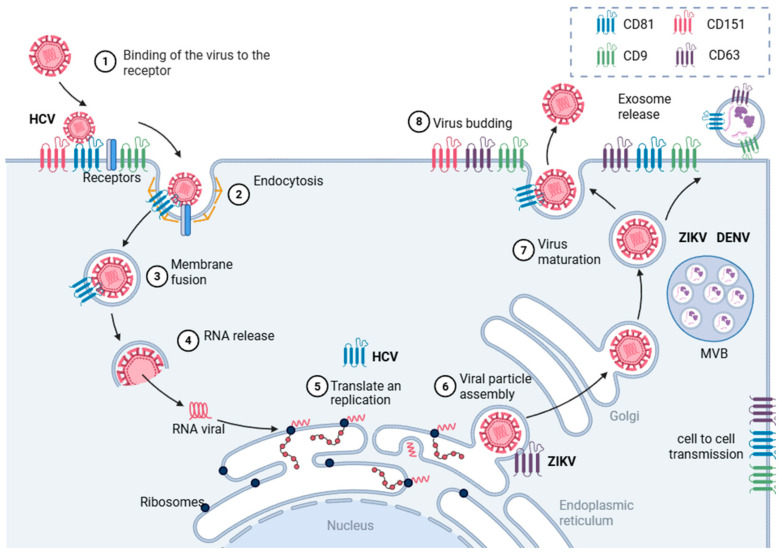
Tetraspanins are involved in several stages of the *Orthoflavivirus* replicative cycle. Tspan participates in viral entry, translation and replication, viral particle assembly, and viral budding or release. *Orthoflavivirus* can also hijack the EVs pathway to enable cell-to-cell, bypassing the conventional entry and egress steps by exploiting Tspan-mediated mechanisms.

**Figure 5 viruses-17-01321-f005:**
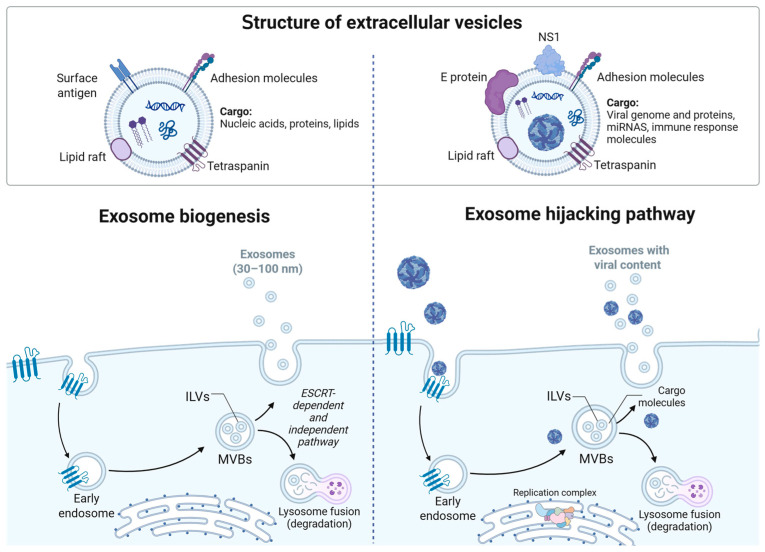
Structure and biogenesis of exosomes and their hijacking by orthoflaviviruses. Exosomes’ biogenesis begins with the formation of early endosomes, which mature into MVBs that contain ILVs formed by ESCRT-dependent or independent pathways. *Orthoflavivirus* exploits this pathway by incorporating viral proteins, genomes, and immune modulators into exosomes. Viral replication complexes interact with the endosomal system, allowing virions or viral components to be secreted within exosomes. This hijacking facilitates viral dissemination, immune evasion, and persistence in the host.

**Table 1 viruses-17-01321-t001:** Tetraspanins’ interaction with cell factors involved in *Orthoflavivirus* entry.

Tetraspanin	Interaction with Cell Factors	Implication in Viral Infection	Reference
CD81	Claudin-1, SR-BI, EGFR, Integrins	Cofactor in HCV, JEV, WNV, DENV, and ZIKV entry	[74,79,80,81,82,83,84,85,86]
CD63	Rab proteins, ESCRT, LAMP-1	Involved in endocytosis and multivesicular body formation	[87,88,89,90,91,92]
CD9	Integrins, EWI-2	Participates in membrane reorganization	[93,94]
CD151	Integrins α3β1, α6β1	Modulates membrane dynamics and signaling during viral entry	[95,96]

**Table 2 viruses-17-01321-t002:** Tetraspanins’ interaction with flavivirus proteins.

Tetraspanin	Viral Protein	Implication in Flavivirus Infection	Reference
CD81	E2-glicoprotein	HCV entry	[67,69]
CD63	Capsid	ZIKV assembly	[98]
Tsp29Fb	Envelope	DENV2 propagation	[99]
CD151	NS2B, capsid	ZIKV and DENV2 Replication and assembly	[102]

**Table 3 viruses-17-01321-t003:** Extracellular vesicle inhibitors with antiviral potential.

Inhibitor	Target	Cell Effect	Reference
GW4869	Neutral sphingomyelinase 2	Inhibits the formation of intraluminal vesicles (ILVs), reducing exosome biogenesis and release.	[153,154]
Imipramine	Acid sphingomyelinase	Inhibits the conversion of acid sphingomyelinase to ceramide and decreases the release of extracellular vesicles.	[155]
DPTIP	Neutral Sphingomyelinase 2	Reduces exosome release.	[156]
Simvastatin	HMG-CoA reductase (cholesterol synthesis)	Decreases membrane cholesterol levels, leading to a significant decrease in exosome secretion	[157]
Indomethacin	ABCA3 (lipid transporter)	Non-selectively inhibits lipid transporter; prevents exosome release.	[154]
Nexinhib20	Rab27A (vesicle trafficking)	Inhibits the fusion of MVBs with the plasma membrane; combined with cisplatin and etoposide, it enhances the inhibitory effect.	[153]
Glibenclamide	ABC transporters	Modulates cholesterol recycling and reduces the release of microvesicles	[158]

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
