# Peer review of "Lipids, Tetraspanins, and Exosomes: Cell Factors in *Orthoflavivirus* Replication and Propagation"

_viruses, 2025, doi:10.3390/v17101321_

Round 1

Reviewer 1 Report

Comments and Suggestions for Authors

This comprehensive and timely review by Benítez-Vega et al. synthesizes the current understanding of the critical roles played by host cell lipids, tetraspanins (Tspan), and extracellular vesicles (EVs) in the flavivirus replication cycle. The topic is highly relevant, given the ongoing global burden of flaviviruses like dengue and Zika and the lack of broadly effective antiviral therapies. The authors effectively argue that these host factors represent promising, underexplored targets for novel therapeutic strategies.

The major strengths of this review lie in its scope and integration of three interconnected cellular systems (lipids, Tspan, EVs) into a cohesive narrative of viral exploitation. The sections on Tspan structure/function and EV-mediated viral dissemination are particularly strong. However, the manuscript requires significant revision to fully realize its potential. The primary concerns are the incomplete presentation (missing clear figures and references) and a need for more critical analysis, especially in the therapeutic section. Addressing the points below will significantly strengthen this valuable contribution to the field.

Major Issues:

Therapeutic Section (Section 9) Requires Depth: This section is currently more descriptive than critical. The authors list various inhibitors (e.g., GW4869, statins) but should provide a more balanced discussion. Key questions to address:

What are the primary challenges of host-directed antiviral therapy (e.g., potential toxicity, off-target effects on essential cellular functions)?

For the inhibitors mentioned, is there any in vivo data (beyond cell culture) demonstrating efficacy against flaviviruses? The mention of limited viral load reduction with DPTIP in vivo is good; similar context for other compounds is needed.

The concept of "broad-spectrum" activity is touted. Could targeting these conserved host pathways potentially lead to resistance? This should be briefly discussed.

Minor Issues:

Language and Flow: The writing is generally clear but would benefit from thorough proofreading by a native English speaker or professional editing service to polish grammar and improve sentence flow in several places. Some sentences are overly long and complex.

Structural Clarity: The transition from Section 3 (TEMs) to Section 4 (TEMs in replicative cycle) is slightly abrupt. Consider merging or adding a stronger concluding sentence to Section 3 that directly leads into the virology focus.

Terminology: The term "Orthoflavivirus" is correctly used in the introduction but later replaced with the more common "flavivirus." For consistency and broader readability, consider using "flavivirus" throughout after the initial taxonomic clarification.

Author Response

Author's Reply to the Review Report (Reviewer 1)

This comprehensive and timely review by Benítez-Vega et al. synthesizes the current understanding of the critical roles played by host cell lipids, tetraspanins (Tspan), and extracellular vesicles (EVs) in the flavivirus replication cycle. The topic is highly relevant, given the ongoing global burden of flaviviruses like dengue and Zika and the lack of broadly effective antiviral therapies. The authors effectively argue that these host factors represent promising, underexplored targets for novel therapeutic strategies.

The major strengths of this review lie in its scope and integration of three interconnected cellular systems (lipids, Tspan, EVs) into a cohesive narrative of viral exploitation. The sections on Tspan structure/function and EV-mediated viral dissemination are particularly strong. However, the manuscript requires significant revision to fully realize its potential. The primary concerns are the incomplete presentation (missing clear figures and references) and a need for more critical analysis, especially in the therapeutic section. Addressing the points below will significantly strengthen this valuable contribution to the field.

Major Issues:

Therapeutic Section (Section 9) Requires Depth: This section is currently more descriptive than critical. The authors list various inhibitors (e.g., GW4869, statins) but should provide a more balanced discussion. Key questions to address:

  1. What are the primary challenges of host-directed antiviral therapy (e.g., potential toxicity, off-target effects on essential cellular functions)?

Response: We thank the reviewer for this valuable suggestion. We fully agree that the potential toxicity and off-target effects of host-directed antivirals remain a significant challenge. Although several compounds (e.g., statins, EV inhibitors) have shown promising antiviral activity, current knowledge about their toxicity and safety profile is still limited, in therapeutic application is constrained by potential host resistance and toxicity. To address this important point, we have revised Section 9 to more clearly highlight these limitations and underscore the need for selective targeting. This is particularly relevant in the case of EV modulation, where distinguishing between proviral and antiviral EV subpopulations is critical. These revisions have been incorporated into the main manuscript as suggested.

  1. For the inhibitors mentioned, is there any in vivo data (beyond cell culture) demonstrating efficacy against flaviviruses? The mention of limited viral load reduction with DPTIP in vivo is good; a similar context for other compounds is needed.

Response: We thank the reviewer for this insightful comment. We have revised the section to better emphasize the available in vivo evidence regarding flavivirus inhibition. Specifically, we expanded the discussion to include studies showing that treatment with statins, ezetimibe, and metformin has been associated with reduced viremia, decreased tissue viral loads, and attenuation of clinical disease severity in murine models. In contrast, the inhibitor GW4869, which blocks neutral sphingomyelinase 2 (nSMase2) and ceramide production, has only been evaluated in adult Ixodes scapularis ticks, where reduced LGTV dissemination was observed. While these findings support its potential role in modulating virus spread, no mammalian in vivo data are available.  Other inhibitors of ceramide metabolism, such as imipramine, have demonstrated antiviral activity against ZIKV, WNV, and DENV in vitro, but likewise lack in vivo validation. Importantly, DPTIP remains the only inhibitor with reported antiviral activity in a murine model, although the observed reduction in viral load was modest. These revisions have been incorporated into the main manuscript to provide a more comprehensive overview of preclinical data.

  1. The concept of "broad-spectrum" activity is touted. Could targeting these conserved host pathways potentially lead to resistance? This should be briefly discussed.

Response: We thank the reviewer for this valuable suggestion. We have added a brief discussion highlighting that host-directed therapies exert pleiotropic effects by modulating multiple signaling pathways. This broad impact generates diverse therapeutic targets and allows for synergistic interactions, which may lower the likelihood of resistance development compared to direct-acting antivirals that rely on single viral targets. These revisions have been incorporated into the main manuscript.

Minor Issues:

Language and Flow: The writing is generally clear but would benefit from thorough proofreading by a native English speaker or professional editing service to polish grammar and improve sentence flow in several places. Some sentences are overly long and complex.

Response: We thank the reviewer for this important observation. We have carefully revised the manuscript to enhance grammar, readability, and sentence flow. Additionally, we obtained external proofreading to ensure the language is polished, and we simplified overly long or complex sentences. These changes have been incorporated into the revised version of the manuscript.

Structural Clarity: The transition from Section 3 (TEMs) to Section 4 (TEMs in replicative cycle) is slightly abrupt. Consider merging or adding a stronger concluding sentence to Section 3 that directly leads into the virology focus.

Response: We appreciate the reviewer’s insightful comment. To improve structural clarity, we have added a concluding sentence at the end of Section 3 that explicitly links the general description of TEMs to their functional roles in the flavivirus replicative cycle, thereby creating a smoother and more logical transition into Section 4. This revision has been incorporated into the main manuscript.

Terminology: The term "Orthoflavivirus" is correctly used in the introduction but later replaced with the more common "flavivirus." For consistency and broader readability, consider using "flavivirus" throughout after the initial taxonomic clarification.

Response: We thank the reviewer for pointing out this inconsistency. To ensure taxonomic accuracy and consistency throughout the manuscript, we have revised the text to use the term “orthoflavivirus” consistently after the initial taxonomic clarification in the Introduction. This change applies to all instances previously labeled as “flavivirus” and has been incorporated into the main manuscript.

Reviewer 2 Report

Comments and Suggestions for Authors

In this decade, many efforts to elucidate functions of extracellular vesicles including exosomes have been made. Along with those research, evidences that flaviviuses utilize EVs to replicate and disseminate cell to cell have been accumulated. This manuscript well reviewed the interactions of lipids, tetraspanin and EVs with flaviviruses. Thus it should be quite interesting to many readers. Only several points were raised.

  1. Line 199-200: Please indicate those are attachment factors to avoid misunderstanding those are receptors.
  2. Table 2: Please add “flavivirus” to table title.
  3. “8. Extracellular vesicles during flavivirus infection”: In this section, many papers showed that E proteins were found in the EVs. Were those E proteins found on the surface of the EVs? Or found inside of the EVs?
  4. Relate to above comment, do EVs containing E protein utilize E-receptor binding to enter cells? It is recommended to prepare a brief description to state how those EVs internalize into cells. Probably those EVs utilize several pathways that EVs generally use.
  5. Line 373: Please remove “specific”, because those drugs may effective to broad viruses.
  6. Table 3: Combination of GW4869 and Nexinhib20 was not assessed in the reference 137.
  7. EVs should contain host-derived miRNAs which induce antiviral effects. Thus to develop drugs inhibiting EV generation as antivirals is considered as controversial. How do authors think about this issue?

Author Response

Author's Reply to the Review Report (Reviewer 2)

In this decade, many efforts to elucidate functions of extracellular vesicles including exosomes have been made. Along with those research, evidences that flaviviuses utilize EVs to replicate and disseminate cell to cell have been accumulated. This manuscript well reviewed the interactions of lipids, tetraspanin and EVs with flaviviruses. Thus it should be quite interesting to many readers. Only several points were raised.

  1. Line 199-200: Please indicate those are attachment factors to avoid misunderstanding those are receptors.
    Response: We thank the reviewer for this important clarification. We have revised the manuscript to indicate that these molecules act as attachment factors, to avoid misunderstanding that they are bona fide receptors. This change has been incorporated into the main manuscript.
  2. Table 2: Please add “flavivirus” to table title.

Response: We thank the reviewer for this suggestion. The title of Table 2 has been updated to include “flavivirus” to ensure clarity and consistency. This modification has been incorporated into the main manuscript.

3. “8. Extracellular vesicles during flavivirus infection”: In this section, many papers showed that E proteins were found in the EVs. Were those E proteins found on the surface of the EVs? Or found inside of the EVs?

Response: We thank the reviewer for raising this important point. In both ZIKV and DENV, virus-like particles have been observed within exosomes by transmission electron microscopy, suggesting a novel transmission route (106, 108). Additionally, studies with ZIKV indicate that the E protein can be present on the surface of exosomes, where it modulates antibody-dependent enhancement mediated by ZIKV E-specific and DENV-cross-reactive antibodies in both cell culture and mouse models. This clarification regarding the localization of viral proteins in EVs has been added to the main manuscript.

4. Relate to above comment, do EVs containing E protein utilize E-receptor binding to enter cells? It is recommended to prepare a brief description to state how those EVs internalize into cells. Probably those EVs utilize several pathways that EVs generally use.

Response: We appreciate the reviewer’s insightful comment. We have added a brief description noting that EVs containing E proteins likely internalize through multiple pathways commonly used by extracellular vesicles, rather than exclusively via classical E-receptor interactions. These pathways may include endocytosis, macropinocytosis, and other cell type–dependent mechanisms. This clarification has been incorporated into the main manuscript to provide a more complete understanding of EV entry processes.

5. Line 373: Please remove “specific”, because those drugs may effective to broad viruses.
Response: We thank the reviewer for this observation. The term “specific” has been removed to accurately reflect that these compounds may exhibit broader antiviral activity.

6. Table 3: Combination of GW4869 and Nexinhib20 was not assessed in the reference 137.

Response: We have corrected Table 3 to accurately reflect the data from reference 137, removing any suggestion that the combination of GW4869 and Nexinhib20 was evaluated. This correction has been incorporated into the main manuscript.

7. EVs should contain host-derived miRNAs which induce antiviral effects. Thus to develop drugs inhibiting EV generation as antivirals is considered as controversial. How do authors think about this issue?

Response: We thank the reviewer for raising this important point. We have added a discussion highlighting the dual role of EVs during infection: while EVs can facilitate viral dissemination, they can also mediate antiviral signaling. For instance, a study reported that EVs from DENV-infected monocyte-derived dendritic cells (mdDCs) contained several mRNAs and miRNAs associated with immune responses compared to EVs from mock-infected mdDCs. These findings suggest that, during DENV infection, the EV pathway can be exploited to support viral replication, yet EVs derived from immune cells may also contribute to antiviral defense mechanisms. These results were incorporated into the main manuscript.

Reviewer 3 Report

Comments and Suggestions for Authors

The full report is attached.

Author Response

This manuscript presents a narrative review exploring how host-derived lipid metabolism, tetraspanin-enriched microdomains, and exosome-mediated intercellular communication contribute to the replication and dissemination of flaviviruses. The authors attempt to integrate recent literature regarding Zikavirus (ZIKV), Dengue virus (DENV), West Nile virus (WNV), and others, focusing on host cell processes that modulate viral replication, immune evasion, and extracellular vesicle (EV)-mediated spread. The topic is timely and relevant, especially considering the increasing attention on host factors in viral pathogenesis and the potential for therapeutic targeting of lipid pathways and exosomal machinery. However, in its current form, the manuscript suffers from terminological inconsistencies, shallow mechanistic discussion, insufficient referencing of recent literature, and some structural disorganization.

  1. Throughout the manuscript, the term “flavivirus” is used inconsistently or Many of the viruses discussed (DENV, ZIKV, WNV) belong to the Orthoflavivirus genus, which should be used explicitly in scientific writing under current ICTV guidelines. Replace "flavivirus" with "orthoflavivirus" throughout when referring to viruses in this genus. If referring more broadly to the Flaviviridae family (e.g., HCV), clarify that distinction.

Response:
 We thank the reviewer for this valuable observation regarding taxonomy. In response, we have thoroughly revised the manuscript to ensure consistent and accurate usage of terminology. Specifically, we now use the term “orthoflavivirus” when referring to members of the Orthoflavivirus genus (e.g., DENV, ZIKV, WNV), following the current ICTV guidelines. In instances where we refer more broadly to the Flaviviridae family (e.g., HCV), we have explicitly clarified this distinction. These modifications have been applied consistently throughout the manuscript to improve clarity and scientific accuracy.

  1. Mechanistic explanations are often surface-level. For example, the roles of PI3K/AKT in lipid droplet formation or the modulation of cholesterol biosynthesis by viral NS proteins are mentioned briefly without discussing supporting or conflicting Important concepts such as autophagy modulation by NS4A/NS4B, or tetraspanin-dependent sorting into multivesicular bodies (MVBs), are glossed over without detail. Expand sections with clearer subheadings and at least one mechanistic figure that illustrates viral modulation of host pathways (e.g., lipid droplet biogenesis or EV loading).

Response: We thank the reviewer for highlighting the need for a deeper mechanistic discussion. We have expanded the relevant sections to provide more detailed explanations. In the revised manuscript, we now describe in greater detail the PI3K/Akt/mTOR pathway, which plays an essential role in lipid metabolism by regulating both catabolic and anabolic processes, and discuss how viral proteins such as NS4A and NS4B modulate these pathways. We have also expanded the section on extracellular vesicles (EVs) to include additional details on the ESCRT-independent biogenesis pathway, particularly the role of tetraspanins in MVB sorting. Furthermore, we have incorporated a new figure that illustrates the modulation of lipid metabolism and EV biogenesis by viral proteins.

  1. The link between tetraspanins (e.g., CD63, CD81) and the exosomal pathway is alluded to but not explored in depth. How orthoflaviviruses hijack or modulate the ESCRT machinery or tetraspanin-enriched microdomains remains vague. Add a dedicated section discussing known interactions between orthoflaviviral proteins and tetraspanins (e.g., ZIKV envelope protein and CD63); role of tetraspanins in exosome biogenesis vs viral particle formation, and key studies that show packaging of viral RNAs/proteins into

Response: We appreciate the reviewer’s insightful comments. In the revised manuscript, we have added a dedicated section (Section 7) that discusses in detail the interactions between orthoflaviviral proteins and tetraspanins. This section addresses the role of tetraspanins in exosome biogenesis and how orthoflaviviruses hijack these processes, with specific reference to interactions such as the ZIKV proteins with CD63. We also highlight key studies demonstrating the packaging of viral RNAs and proteins into exosomes and discuss the functional implications of these findings in the context of viral replication and dissemination. These additions provide greater depth and clarity to the manuscript.

  1. The manuscript would benefit from a clearer structure. Currently, it transitions unpredictably between viruses and cell components without consistent framing. My suggestion is:
    • Introduction
    • Host lipid metabolism and viral replication
    • Tetraspanin microdomains in viral entry and egress
    • Exosomes and intercellular viral spread
    • Therapeutic opportunities and conclusions

Response: We sincerely thank the reviewer for this thoughtful suggestion regarding the manuscript’s structure. While we agree that the proposed outline is clear and well             organized, we respectfully prefer to maintain the current structure, as our intention was to       emphasize the central role of tetraspanins throughout the review. This approach allows us      to integrate their interplay with lipids, exosomes, and viral processes in a more                               interconnected manner rather than treating these topics in separate, isolated sections.

That said, we have carefully revised the transitions between sections to improve clarity           and coherence, ensuring a smoother flow and more consistent framing of the discussion       across viruses and host cell components.

  1. The English is mostly clear, but some sentences are overly long or contain unnecessary filler words. Example: “These data collectively show that viruses such as ZIKV and DENV have evolved to manipulate cellular processes...” --> “These findings show that orthoflaviviruses can modulate host cellular ..”

Response: We thank the reviewer for this valuable observation. We have carefully revised the manuscript to improve grammar, readability, and sentence flow. Overly long or complex sentences have been simplified, and unnecessary filler words have been removed. Additionally, the revised version of the manuscript has undergone external proofreading to ensure clarity and polished language throughout.

This manuscript is built on a promising and relevant idea but requires major revisions to meet the standards of a high-quality review. With more rigorous citation, updated taxonomy, improved structure, and deeper mechanistic exploration, it has the potential to contribute meaningfully to the literature on orthoflavivirus—host interactions.

Response. We sincerely thank the reviewer for their overall assessment and constructive feedback. We have addressed each of the concerns by incorporating updated taxonomy in line with ICTV guidelines, expanding mechanistic explanations with additional references and details, and adding a new figure to illustrate viral modulation of host pathways. Furthermore, we revised the manuscript to strengthen citation of recent and relevant studies, improve section transitions for greater clarity, and include a dedicated section on tetraspanins and exosome-mediated processes. We believe that these substantial revisions have significantly enhanced the depth, accuracy, and overall quality of the review, and we hope the revised version now meets the standards of a high-quality contribution to the literature on orthoflavivirus–host interactions.

Round 2

Reviewer 2 Report

Comments and Suggestions for Authors

Authors revised well based on my comments. Thus, current manuscript is now considered as acceptable in "Viruses". This review article should be quite informative to many researchers.

Author Response

Authors revised well based on my comments. Thus, current manuscript is now considered as acceptable in "Viruses". This review article should be quite informative to many researchers.

Response: 

We sincerely thank the reviewer for their positive and constructive evaluation. We are pleased that the revisions have improved the review's structure, taxonomic accuracy, and mechanistic depth. Following the reviewer's suggestions, we carefully incorporated changes throughout the manuscript to further enhance readability. We are confident that the revised version will provide readers with a more precise and comprehensive synthesis.

Reviewer 3 Report

Comments and Suggestions for Authors

The report is attached.

Comments on the Quality of English Language

The manuscript is written in generally clear and readable English, with substantial improvements over the original version. However, a few stylistic refinements would further polish the paper:
-Some sentences are still overly long or include redundant connectors (“this thus suggests that…”).
-Phrasing can occasionally be tightened for fluency, e.g., "it is important to note that orthoflaviviruses..." => "orthoflaviviruses notably..."
-Subject–verb agreement and article usage have been corrected in most places, but one final proofreading by a native or fluent English speaker is advised to improve naturalness.

Author Response

15 September 2025  

Second report manuscript "Lipids, tetraspanins, and exosomes: cell factors in flavivirus replication and propagation"  

Journal Viruses - MDPI  

This revised narrative review examines how host cellular components (specifically lipid metabolism, tetraspanin-enriched microdomains, and exosome-mediated intercellular communication) are co-opted by orthoflaviviruses (such as DENV, ZIKV, and WNV) to enhance replication, immune evasion, and cell-to-cell spread. The authors have addressed previous reviewer concerns by correcting terminological usage (notably "orthoflavivirus"), deepening mechanistic discussions, revising sentence structure, and adding a new illustrative figure and a section on tetraspanin-viral interactions. The review is now better structured, more taxonomically accurate, and provides a more insightful discussion into key host-virus interactions. Overall, it contributes a relevant and timely synthesis to the field of arbovirology and cellular virology.  

  1. The manuscript now provides improved mechanistic explanations of key pathways, such as PI3K/AKT/mTOR in lipid droplet metabolism and ESCRT-independent exosome biogenesis. The new dedicated section on tetraspanins and orthoflavivirus interactions is particularly valuable, as this is an underrepresented area in review literature. The topic is timely and contributes to our understanding of host factors influencing viral pathogenesis and potential antiviral targets.  

Response: We sincerely thank the reviewer for their positive and constructive evaluation. We are pleased the revisions have improved the review's structure, taxonomic accuracy, and mechanistic depth. Following the reviewer's suggestions, we carefully incorporated changes throughout the manuscript to further enhance readability and ensure the figure's legibility. We appreciate the recognition of the added section on tetraspanin–orthoflavivirus interactions, which we agree represents an underexplored but important area in the field. We are confident that the revised version will provide readers with a more precise and comprehensive synthesis. 

  1. Although the authors did not adopt the suggested restructuring (i.e., separating topics into discrete sections), the internal coherence of their preferred framework is now much improved. Transitions between sections are smoother, and subheadings are clearer. Figure 2 meaningfully aids comprehension of the host-pathogen interface.  

Response: We are pleased that the revised organization was found to be more internally consistent, even though we maintained the original framework. We sincerely appreciate the reviewer's recognition of the improved transitions and subheadings and the positive feedback on Figure 2. This figure was designed to strengthen the integration of host–pathogen interactions and to provide readers with a more intuitive understanding of the concepts discussed. 

  1. "Orthoflavivirus" is now correctly and consistently used, following ICTV guidelines. The distinction between Orthoflavivirus and other Flaviviridae members is clear and contextually appropriate. 

Response: We thank the reviewer for this positive remark. In revising the manuscript, we carefully ensured the consistent use of the term Orthoflavivirus in accordance with ICTV guidelines, and we emphasized its distinction from other members of the Flaviviridae family. We are pleased that these clarifications have improved the accuracy and contextual appropriateness. 

  1. Recent and relevant literature has been incorporated to support mechanistic claims. Several high-impact studies (e.g., on NS4A/NS4B and tetraspanin interactions) have now been included.  
    Response: We thank the reviewer for recognizing the incorporation of recent and high-impact studies. Our goal was to reinforce the mechanistic perspective of the review, and we are pleased that the additional references—particularly those addressing NS4A/NS4B and tetraspanin interactions—were considered valuable in enhancing the scientific depth of the manuscript. 
  1. The new figure is conceptually helpful, though it may benefit from slightly larger font and clearer distinction between host and viral components. Consider including a brief caption legend summary of the figure's relevance for ease of interpretation.  

Response: We thank the reviewer for this practical suggestion. In response, we increased the font size and enhanced visual contrast between host and viral components by using distinct shapes and clearer labeling. Additionally, we expanded the figure legend to provide a concise summary of the figure's purpose and its relevance to the host–pathogen interface. The revised figure and legend are now included in the main manuscript. 

  1. Line 68-72: consider tightening the introduction here for clarity. The term "unified model" is used, but no such model is ultimately proposed: rephrase or elaborate.  
    Response: We appreciate the reviewer's observation. To avoid confusion, we revised the introduction by removing the term "unified model" and rephrasing the section for greater clarity. These changes have been incorporated into the main manuscript. 
  1. Line 155-168 (PI3K/AKT pathway): this expansion is helpful. You may wish to cite one or two additional studies that directly show how NS4A or NS4B manipulate this pathway in DENV/ZIKV.  
    Response: We thank the reviewer for this valuable recommendation. In response, we have incorporated citations to recent studies that directly document the manipulation of PI3K/AKT signalling by NS4A/NS4B in flaviviruses, including JEV. To further strengthen the mechanistic discussion, we added a summary of the key experimental findings from these reports. These additions have been included in the revised text at lines ~90–93 of the main manuscript. 
  1. Line 223-238 (Autophagy): the section has improved, but a brief distinction between canonical vs non-canonical autophagy could provide context, especially when discussing lipid droplet degradation.  
    Response: We appreciate this constructive suggestion. In the revised manuscript, we inserted a brief explanation distinguishing canonical from non-canonical autophagy pathways, highlighting their different implications for lipid droplet degradation and the exploitation of host lipophagy by orthoflaviviruses. To support this addition, we also included an appropriate reference. These clarifications have been incorporated into the main text at lines ~96–115. 
  1. Section 7 (Tetraspanins): excellent addition. Please double-check that all abbreviations (e.g., MVBs, ESCRT, EVs) are clearly defined when first used.  
    Response: We thank the reviewer for this helpful editorial observation. We carefully reviewed Section 7 and the entire manuscript to ensure that all abbreviations (e.g., MVBs, ESCRT, EVs) are clearly defined at first mention and used consistently throughout the text. These editorial corrections have been implemented in the revised manuscript. 
  1. Lines 317-322: the discussion of therapeutic implications is thoughtful, but still speculative. Consider tempering claims such as "targeting host lipid pathways may inhibit viral spread" to more cautious phrasing, e.g., "represent a potential therapeutic avenue requiring further validation".  
    Response: We agree with the reviewer's thoughtful observation. We have revised the relevant sentences (lines 436–438 and adjacent text) to adopt more cautious and balanced phrasing, indicating that targeting host lipid pathways represents a potential therapeutic avenue that requires further validation. These changes have been incorporated into the revised manuscript. 

The authors have significantly improved the manuscript in response to prior reviewer critiques. The paper is now suitable for publication in Viruses, pending very minor language refinements and optional figure enhancements.